

# Regression assumptions in clinical psychology research practice—a systematic review of common misconceptions

Anja F. Ernst and  Casper J. Albers

Heymans Institute for Psychological Research, University of Groningen, Groningen, The Netherlands

## ABSTRACT

Misconceptions about the assumptions behind the standard linear regression model are widespread and dangerous. These lead to using linear regression when inappropriate, and to employing alternative procedures with less statistical power when unnecessary. Our systematic literature review investigated employment and reporting of assumption checks in twelve clinical psychology journals. Findings indicate that normality of the variables themselves, rather than of the errors, was wrongfully held for a necessary assumption in 4% of papers that use regression. Furthermore, 92% of all papers using linear regression were unclear about their assumption checks, violating APA-recommendations. This paper appeals for a heightened awareness for and increased transparency in the reporting of statistical assumption checking.

# INTRODUCTION

One of the most frequently employed models to express the influence of several predictors on a continuous outcome variable is the linear regression model:

$$Y_i = \beta_0 + \beta_1 X_{1i} + \beta_2 X_{2i} + \cdots + \beta_p X_{pi} + \varepsilon_i.$$

This equation predicts the value of a case $Y_i$ with values $X_{ji}$ on the independent variables $X_j$ ($j = 1, \ldots, p$). The standard regression model takes $X_j$ to be measured without error (cf. *Montgomery, Peck & Vining, 2012*, p. 71). The various $\beta_j$ slopes are each a measure of association between the respective independent variable $X_j$ and the dependent variable $Y$. The error for the given $Y_i$, the difference between the observed value and value predicted by the population regression model, is denoted by $\varepsilon_i$ and is supposed to be unrelated to the values of $X_p$. Here, $\beta_0$ denotes the intercept, the expected $Y$ value when all predictors are equal to zero. The model includes $p$ predictor variables. In case $p = 1$, the model is denoted as the simple linear regression model.

The standard linear regression model is based on four assumptions. These postulate the properties that the variables should have in the population. The regression model only provides proper inference if the assumptions hold true (although the model is robust

Corresponding author
Casper J. Albers, c.j.albers@rug.nl

to mild violations of these assumptions). Many statistical textbooks (for instance, *Miles & Shevlin, 2001*; *Cohen et al., 2003*; *Lomax & Hahs-Vaughn, 2012*; *Montgomery, Peck & Vining, 2012*) provide more background on these assumptions as well as advice on what to do when these assumptions are violated.

Violations of these assumptions can lead to various types of problematic situations. First, estimates may become biased, that is not estimating the true value on average. Second, estimators may become inconsistent, implying that convergence to the true value when the sample size increases is not guaranteed. Third, the ordinary least squares estimator may not be efficient anymore: For instance, in the presence of assumption violations, OLS may provide less accurate parameter estimates than other available estimation procedures. Fourth and finally, NHST's and confidence intervals might become untrustworthy: *p*-values can be systematically too small or too large, and confidence intervals are too narrow or too wide. This can occur even if estimators are unbiased, consistent and efficient. For a more detailed description of these issues, see *Williams, Grajales & Kurkiewicz (2013)*. Please note that these assumptions are the assumptions when estimating using the Ordinary Least Squares (OLS) procedure, which is the default procedure in many software packages, including SPSS and *R*. Other type of estimation methods, such as GLS, apply other sets of assumptions.

Below, the four OLS-assumptions will be discussed.

*Linearity.* The conditional mean of the errors is assumed to be zero for any given combination of values of the predictor variables. This implies that, for standard multiple regression models, the relationship between every independent variable $X_i$ and the population mean of the dependent variable $Y$, denoted by $\mu_Y$, is assumed to be linear when the other variables are held constant. Furthermore, the relations between the various $X_i$ and $\mu_Y$ are additive: thus, the relation of $X_i$ with $\mu_Y$ is the same, regardless of the value of $X_j$ ($j \neq i$). This relates to the issue of multicollinearity; a good model is expected to have as little overlap between predictors as possible. However, multicollinearity is not a model assumption but merely a necessity for a model to be parsimonious. Violation of this assumption can obviously occur when non-linear relations are unmodelled, but also in case of measurement error (see *Williams, Grajales & Kurkiewicz, 2013*).

*Normality.* All errors are normally distributed around zero.

*Homoscedasticity.* The variance of the errors is the same for any combination of values of the independent variables. Thus, this variance, which can then be denoted by a single symbol (e.g., $\sigma^2$). This assumption is also called the homoscedasticity assumption. Thus, the second and third regression assumptions combined specify that the errors ($\varepsilon_i$) of the model should follow a normal distribution with a mean of zero and a (fixed) standard deviation $\sigma$. Heteroscedasticity often manifests itself through a larger spread of measurements around the regression line at one side of the scatterplot than at the other.

*Independence.* The errors $\varepsilon_1, \varepsilon_2, \ldots$, should be independent of one another: the pairwise covariances should be zero. This assumption is not directly based on the distribution of the data but on the study design and it requires the sampling method to be truly random (see, for instance, *Cohen et al., 2003*). As with the normality assumption, scatterplots alone
are usually unsuitable for checking this assumption. A residual plot, or inspection of the autocorrelation of the residuals, is a better approach.

*Common misconceptions about assumptions.* There are many misconceptions about the regression model, most of which concern the assumptions of normality and homoscedasticity. Most commonly, researchers incorrectly assume that $X_i$, or both $X_i$ and $Y$, should be normally distributed, rather than the errors of the model. This mistake was even made in a widely-read article by *Osborne & Waters (2002)* attempting to educate about regression assumptions[1] (cf. *Williams, Grajales & Kurkiewicz, 2013*).

Not assuming a normal distribution for $X_i$ may seem counterintuitive at first, however the indulgence of this assumption becomes more evident with an illustrative example. Take the standard Student's $t$-test which assesses if two distributions are statistically different from one another (e.g., a $t$-test that compares the efficacy of a specific treatment compared to a placebo treatment). The population distributions in both conditions are assumed to be normally distributed with equal variances. This $t$-test can also be expressed as a regression model where the independent variable $X$ dummy codes the group membership, (i.e., if a participant is in the control $= 0$, or in the treatment condition, $X = 1$). This regression model and the $t$-test are mathematically equivalent and will thus lead to identical inference. Variable $X$ will only attain two values, 0 and 1, as it is only used as label for group membership. The dependent variable $Y$ will attain many different values: following a normal distribution for the treatment group and a (possibly other) normal distribution for the control group. This resulting 'condition membership' distribution is nothing close to normal (as it takes on just two values), however no assumption of the general linear model is violated because the *subpopulations* of $Y$ for each of the $X$ values follow a normal distribution with equal variances, as is visualised in Fig. 1. This example demonstrates that the assumptions of the $t$-test (standard normal distribution of the populations around the group mean and equal variances) coincide with the second regression assumption.

As a consequence of the second regression assumption, the distribution of the dependent variable conditional on some combination of values of the predictor variables is linear. Thus, $Y_i$ is actually normally distributed around $\mu_Y$, the true conditional population mean. This becomes clear when remembering that the error of the regression estimation is normally distributed around mean zero and that $Y_i$ is equal to $\mu_Y + \varepsilon_i$. That is, individual observations are the sum of the mean and a deviation from this mean. However, it is wrong to test the normality of the marginal distribution of the dependent variable $Y$ because this would imply that all $\mu_Y$ values are the same which is, generally, not the case. (This situation occurs only when all regression slopes are zero and, thus, all predictor variables are linearly unrelated to $Y$.)

Regarding the linearity assumption, a common misconception is in thinking that only linear relationships can be modelled using the OLS framework. This is not the case: the linearity assumption deals with linearity in the parameters and the estimates, but not necessarily in the variables.

*Consequences of violations of assumptions.* Misconceptions like the ones outlined above potentially have severe effects on the ability to draw inferences from a data analysis. First of all, the checking of wrong assumptions will most likely lead to the neglect of correct

[1] Based on the journal's access counter, there were more than 540,000 views at the time of this writing (http://pareonline.net/genpare.asp?wh=0&abt=8).

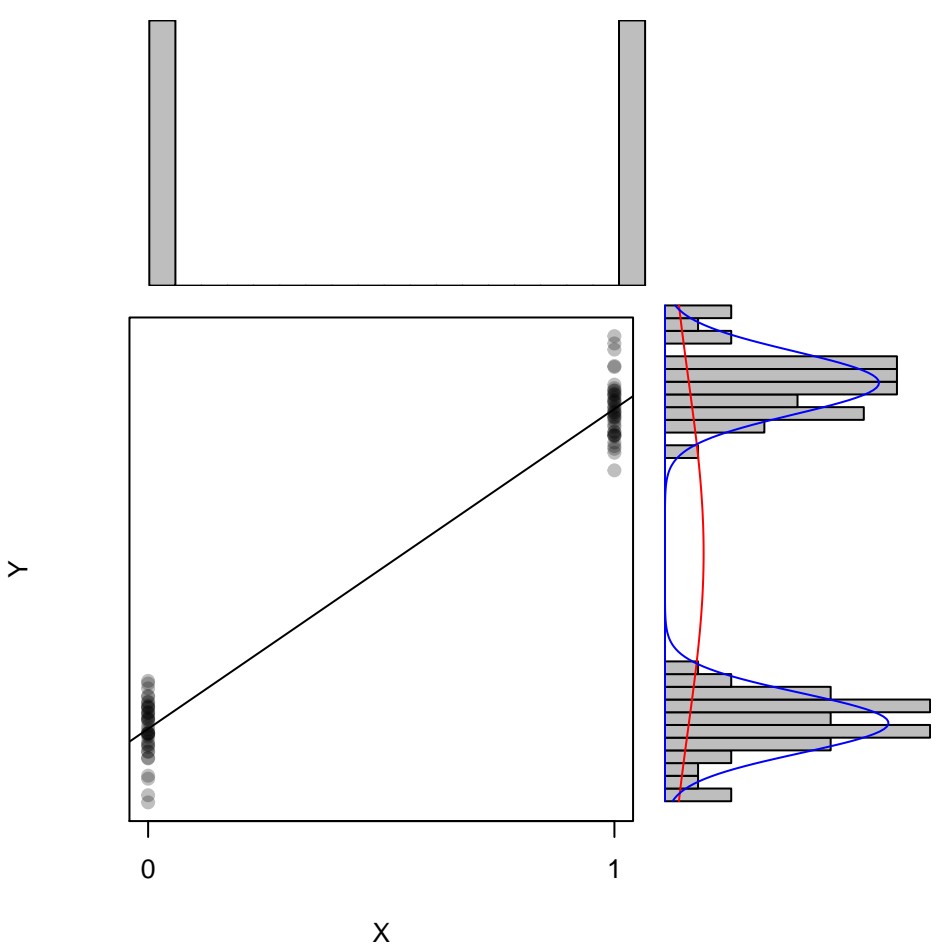

**Figure 1** **Simulated example of a *t*-test based on *n* = 40 observations per group and no violations of the assumptions.** The main panel shows a scatterplot of (*X*, *Y*)-scores. The red curve corresponds to the best-fitting normal distribution for *Y*, where the blue curves correspond to the best-fitting normal distribution for both subpopulations of *Y*. The histograms in the top and side panels clearly indicate non-normality for *X* and *Y*. However, within both subpopulations the distribution is normal.

assumption checking. If the researcher will decide on a regression analysis without having tested the correct assumptions it is possible that some requirements of linear regression were not met. However, in any case the neglect of correct assumption checking will always leave the reader or reviewer unable to trust the results because there is no way of knowing whether the model assumptions were actually met. Of course, the severity of this problem of non-transparency persists even when the researcher ensured the validity of all necessary assumptions and merely failed to report those findings. Not only does such non-transparency in data analysis lead to confusion for researchers that are potentially interested in replicating or comparing the results, it also weakens the informational value of the research findings that are being interpreted.

A second problem that is caused by misconceptions about model assumptions occurs when a researcher decides against a linear regression analysis because of the violation of faulty assumptions that were unnecessary in the first place. The difficulty of abandoning

linear regression analysis for a non-parametric procedure is the fact that the ordinary least squares method of linear regression is a more powerful procedure than any of its non-parametric counterparts, if its assumptions are met. Hence, wrongfully deciding against the employment of linear regression in a data analysis will lead to a decrease in power. Thus, the understanding of the correct regression assumptions is crucial because it prevents the abandonment of the linear regression technique in cases in which it would be unjustified. Furthermore, the checking of assumptions has another advantage: it might help the researcher to think about conceptually alternative models. For instance, heteroscedasticity in the data could be a sign of an interaction between one or more of the included independent variables and an independent variable not (yet) included in the model.

Applying a linear regression model when assumptions are violated can lead to (severe) problems, but this does not have to be the case, depending on the type of violation. Violations of the linearity assumption and of the independence assumption can lead to biased, inconsistent and inefficient estimates (*Chatterjee & Hadi, 2006*; *Williams, Grajales & Kurkiewicz, 2013*). A proper check on these two assumptions is thus vital. The consequences of violations are less severe for the other two assumptions.

If normality of errors holds, the OLS method is the most efficient unbiased estimation procedure (*White & MacDonald, 1980*). If this assumption does not hold (but the remaining assumptions do), OLS is only most efficient in the class of linear estimators (see *Williams, Grajales & Kurkiewicz, 2013*, for a detailed discussion). This implies that, as long as the other assumptions are met, estimates will still be unbiased and consistent in the presence of a normality violation, but the $p$-values might be biased. Furthermore, the central limit theorem implies that for large samples the sampling distribution of the parameters will be at least approximately normal, even if the distribution of the errors is not. Hence, the regression model is robust with respect to violations of the normality assumption. Potential problems will, in practice, primarily occur for inferential problems (such as confidence intervals and testing) with small samples.

Similarly, violations of the homoscedasticity assumption are not necessarily problematic. Provided that the very mild assumption of finite variance holds, estimates will still be unbiased and consistent (*Chatterjee & Hadi, 2006*).

*Best practices for checking of assumptions.* There are many different ways to check the four assumptions of the regression model and there generally is no 'uniformly optimal' approach.

Generally, there are two classes of approaches: (i) formal tests (of the style '$H_0$: the assumption is true' vs '$H_A$: the assumption is violated') and (ii) graphical methods. For the normality assumption alone, there is an abundance of formal tests, such as the Shapiro–Wilk test, the Anderson-Darling test and the Kolmogorov–Smirnov test. Which approach is most powerful depends on the kind of violation from normality (*Razali & Wah, 2011*). However, the use of formal tests is discouraged by some (*Albers, Boon & Kallenberg, 2000*; *Albers, Boon & Kallenberg, 2001*). When the normality assumption holds, the null hypothesis of normality will still be rejected in $\alpha$ (usually 5%) of cases. This distorts the $p$-value distribution of the estimates of the regression model, even when no assumptions

are violated. Furthermore, tests for normality only have adequate power in case of large sample sizes. However, when the sample size is large, the central limit theorem implies that violations of normality have only limited effect on the accuracy of the estimates.

Applying graphical methods is therefore a preferred approach. This is also suggested by the statistical guidelines for the APA set up by *Wilkinson & Task Force on Statistical Inference* (*1999*, p. 598): "Do not use distributional tests and statistical indices of shape (e.g., skewness, kurtosis) as a substitute for examining your residuals graphically". This advice builds upon the adagium by *Chambers et al.*, (*1983*, p. 1) that "there is no single statistical tool that is as powerful as a well-chosen graph". A graph simply provides more information on an assumption than a single *p*-value ever can (see also *Chatterjee & Hadi, 2006*, Ch. 4).

The linearity assumption can easily be checked using scatterplots or residual plots: plots of the residuals vs. either the predicted values of the dependent variable or against (one of) the independent variable(s). Note that residuals are the differences between the observed values and the values predicted by the *sample* regression model, whereas errors denote the difference with the values predicted by the *population* regression model. Residual plots are also the best visual check for homoscedasticity. For the normality assumption, it is difficult to judge on the basis of a scatterplot whether the assumption is violated. A histogram of the residuals is also a poor visual check, as the 'shape' of the histogram heavily depends on the arbitrary choice of the bin width, especially in small samples. Normal probability plots, or QQ-plots, provide a much better way to check normality. Finally, a check on the independence assumption is done by studying the autocorrelation function of the residuals. Note that this latter check does check for temporal dependence violations of the independence assumptions, but not for other possible violations such as clustering of observations. Furthermore, a common violation of independence involves repeated-measures designs in which each individual contributes a set of correlated responses to the data because of individual differences.

*Outline of this paper.* Misconceptions about frequently employed statistical tools, like the *p*-value, are not rare, even amongst researchers (see *Bakker & Wicherts, 2011*; *Hoekstra et al., 2014*). Our paper aims to shed light onto potential misconceptions researchers and reviewers might hold about the linear regression model. Therefore, the documentary practices of psychological research papers with the linear regression model and its assumptions were investigated by means of a literature review. In this review, we investigate the proportion of papers where misconceptions around the assumptions of the statistical regression model occurred and which type of misconceptions occurred most often. This will provide important information, as the first step in solving flawed methodology in research is finding out where the flaws are and how predominant they are.

Although the consequences of incorrectly dealing with assumptions can be severe, the APA manual (*American Psychological Association, 2010*) barely provides guidelines on what to report and how to report. It does *recommend* being specific about "information concerning problems with statistical assumptions and/or data distributions that could affect the validity of findings" (p. 248) as part of the Journal Article Reporting Standards, but this is not obligatory. The APA Task Force on Statistical Inference (*Wilkinson & Task Force*

**Table 1** **Selection of Clinical Psychology Journals.** The first column gives the ranking of the journal, the first number denoting the quartile in which the journal falls, the second number the rank of the journal within that quartile.

| Label | Journal |
| --- | --- |
| Q1.1 | Annual Review of Clinical Psychology |
| Q1.2 | Clinical Psychology Review |
| Q1.3 | Journal of Consulting and Clinical Psychology |
| Q2.1 | International Psychogeriatrics |
| Q2.2 | Journal of Attention Disorders |
| Q2.3 | American Journal of Drug and Alcohol Abuse |
| Q3.1 | Zeitschrift fur Klinische Psychologie und Psychotherapie |
| Q3.2 | Journal of Obsessive-Compulsive and Related Disorders |
| Q3.3 | International Journal of Psychology and Psychological Therapy |
| Q4.1 | Internet Journal of Mental Health |
| Q4.2 | Indian Journal of Psychological Medicine |
| Q4.3 | Behaviour Change |

on Statistical Inference, 1999) is more explicit in their recommendations: "You should take efforts to assure that the underlying assumptions required for the analysis are reasonable given the data. Examine the residuals carefully." (p. 598).

In this manuscript we present the findings of our literature review. Because the whole field of psychological science is too broad to study in a single paper, we restrict ourselves to the field of clinical psychology. We investigate how statistical assumptions were covered in various journals of clinical psychology and what types of misconceptions and mistakes are occurring most often. In the discussion section, possible explanations for the reported findings will be offered. The paper will conclude with several proposals of how potential shortcomings in the current practices with linear regression analysis could be overcome.

## METHOD

### Journals

The literature review restricted itself to articles that were published in clinical psychology journals in the year 2013. It is possible that problems with the checking of assumptions are less (or more) prominent in journals with a high impact, which is why we aimed for a selection of journals with varied impact factors. We employed the Scientific Journal Rankings (SJR) as reported on 16 December 2014 by the SCImago Journal and Country Rank (SCImago, 2007) for clinical psychology journals of the year 2013 to divide all clinical psychology journals into four quartiles (Q1–Q4), where Q1 contains the 25% of journals with the highest journal rank, etcetera. From every quartile the three highest ranked journals were selected to be included in the review. Hence, we obtained a balanced selection from all clinical psychology journals, as listed in Table 1. All articles published in the selected journals in 2013 were included, including also papers that had potentially been published online earlier. Letters, journal corrigenda, editorial board articles and book reviews were not included in the review. Basically, articles that were by design not containing a method

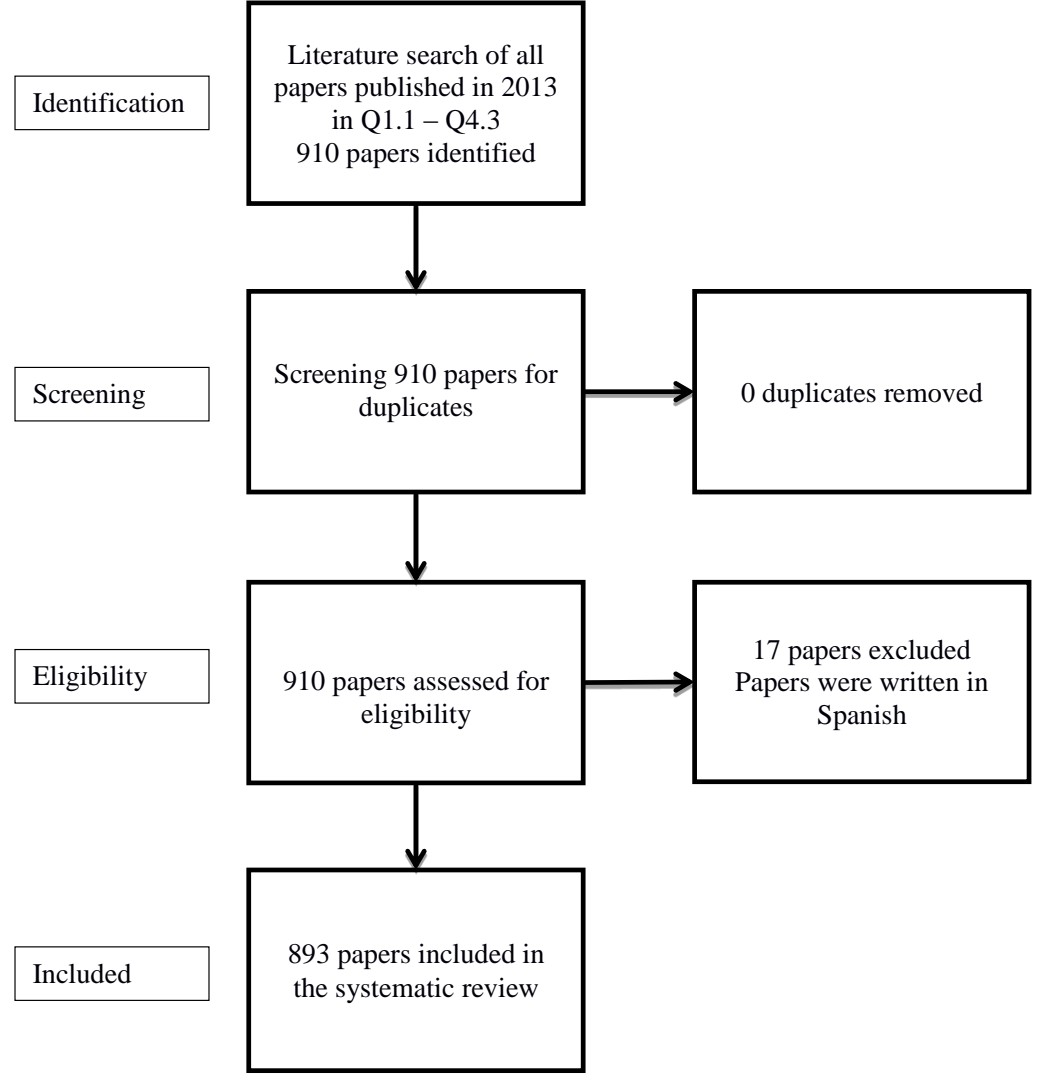

**Figure 2  Prisma flow diagram of included records.**

section were not included in our lists of articles. The focus of this review purely lies on published scientific articles.

Every article was retrieved directly from the official website of its respective journal (except for Q1.3 which was directly retrieved from its official database "PsycARTICLES"). All articles were in German (Q3.1), Spanish (part of Q3.3) or in English (all other). German articles were also included in the review; Spanish articles were excluded because of the authors' lack of proficiency in this language. Figure 2 displays the Prisma workflow of the analysis. We conducted our review adhering to common meta-regression guidelines (*Moher et al., 2009*).

**Table 2   Classification of the reviewed regression papers.** Rubrics 3 and 5–12 represent papers with imperfect handling of regression assumptions: in rubrics 5–7, it is unclear from whether assumptions are correctly dealt with; in rubrics 8–12, the dealing with assumptions was incorrect.

| Class | Reason |
|---|---|
| *Papers without a linear regression model:* | |
| 1 | No Model of Interest |
| 2 | Rejection of linear regression on basis of correct assumptions |
| 3 | Rejection of linear regression on basis of not meeting incorrect assumptions |
| *Papers with a linear regression model:* | |
| 4 | Correct linear regression |
| 5 | Mentioned all correct assumptions but not if the 'normality assumption' was tested on the residuals or on $X$ or $Y$ |
| 6 | Did not test all but some correct assumptions, included neither normality of variables nor errors |
| 7 | Use of linear regression but no indication if any or which assumptions were tested |
| 8 | Assumed/tested normally distributed $X$ but not the normality of the errors |
| 9 | Assumed/tested normally distributed $Y$ but not the normality of the errors |
| 10 | Assumed/tested normally distributed $X$ and $Y$ but not the normality of the errors |
| 11 | Assumed/tested normally distributed variables but did not indicate if $X$ or $Y$ or both and did not test the normality of the errors |
| 12 | Other misconceptions about assumptions |

## Procedure

We evaluated whether and how papers described careful examination of the data with regard to the underlying model assumptions whenever conducting statistical analysis (*American Psychological Association, 2010*; *Wilkinson & Task Force on Statistical Inference, 1999*). Papers were skimmed for the following criteria: if they had used linear regression, how they tested the regression assumptions or what kind of assumptions they indicated as being necessary, if they had transformed data on basis of correct or incorrect assumptions and if a paper had considered an ordinary least squares regression model but employed a different model on basis of either correct or incorrect assumptions. This resulted in a classification scheme of 12 different rubrics which are displayed in Table 2. This scheme is mutually exclusive and exhaustive; all studied papers are classified into exactly one rubric. The search strategy has been carried out by Anja Ernst. Independently, Casper Albers checked and classified 10% of the manuscripts in the Q1-journals. No mismatch between both sets of classifications occurred.

Papers that used linear regression were classified as follows. We assumed the most common misconception about linear regression to be the checking of the normality of the variables while failing to check the normality of the errors. Therefore, we created rubrics

8 to 11 to classify all papers that employed linear regression and checked or assumed the normality of $X$ and/or $Y$ but not of the errors. An example of a paper classified in rubric 8 stated "Variable distributions were tested to ensure assumptions of normality, linearity, and variance equality were met, with no significant violations observed" (*Nadeau et al., 2013*). Often, when the normality assumption was mentioned it was unclear whether authors had checked the normality of errors or of the variables. Articles that were unclear in this regard were classified under rubric 5. For instance, one of the articles classified in this rubric stated "Preliminary analysis examined data for the presence of outliers and the appropriateness of assumptions of normality, linearity, and homoscedasticity" (*Nguyen et al., 2013*) with no more information provided on the assumption checks. Papers that indicated to have checked the homoscedasticity, normality of the errors and linearity assumptions were classified as '*Correct*' in rubric 4. Articles that mentioned at least a few correct assumptions, as opposed to giving no indication at all (rubric 7), were classified in rubric 6. Because all papers that checked or assumed the normality of $X$ or $Y$ but not of the errors were included in rubrics 8 to 11, we have named rubric 6 '*Did not test all but some correct assumptions, did not include normality of variables*'. After performing the literature review it became apparent that none of the articles listed in this category had mentioned the normality of errors. Because we aimed to demonstrate how rare it is to read that researchers check the normality of the errors we have updated the name of the category into '*Did not test all but some correct assumptions, included neither normality of variables nor errors*', even though the checking of the normality of errors was not employed as a criterion for inclusion in this category during the literature review.

Papers that did not fit into any of the eleven other rubrics but included an aspect on linear regression assumptions that we found unsatisfactory were listed in the rubric '*Other misconceptions about assumptions*'. One example of a paper classified in this category claimed "All assumptions of multiple regression (linearity, multicollinearity, and homoscedasticity) were met" this paper was included in the category '*Other misconceptions*' because they did not only lack any mention whether normality of the residuals was checked (which would have resulted in a classification in rubric 6) but also claimed that a list not containing normality of residuals was complete. We found this claim unsatisfactory which was the reason we included this paper in rubric 12.

Whenever an article in our selection reported the results of a regression analysis of another paper or reviewed several linear regression articles, it was evaluated whether the paper reviewing all the previous regression analysis had made it a criterion of inclusion whether the assumptions have been met in the original articles. If a review article did not check or mention the assumptions of the papers that published the original analysis, the article was classified as '*Use of linear regression but no indication if any or which assumptions were tested*'. However, these sorts of papers constitute less than one percent of our selected articles. It should be noted that this only applies to papers which reported the data values of a linear regression or analysed regression results from other studies. A paper was not included if it only mentioned the direction of the outcomes of another paper's regression model or stated that a relationship had been established by previous research findings.

**Table 3 Proportion of various types of papers in our selected journals.** Categorisations are mutually exclusive and exhaustive. Journals are referred by the labels assigned in Table 1. "Rub." refers to the rubrics in Table 2. The online Supplemental Information 1 indicates which papers belong to each of the numbers in this table.

| Journal | Number of papers (rub. 1–12) | Number of papers with regression (rub. 4–12) | Dealing with assumptions | | | No regression | |
|---|---|---|---|---|---|---|---|
| | | | Correctly (rub. 4) | Unclear (rub. 5–7) | Incorrectly (rub. 8–12) | Correct (violation of true assumption) (rub. 2) | Incorrect (violation of false assumption) (rub. 3) |
| Q1.1 | 33 | 0 | 0 | 0 | 0 | 0 | 0 |
| Q1.2 | 86 | 6 (7%) | 0 | 6 (100%) | 0 | 0 | 0 |
| Q1.3 | 98 | 26 (28%) | 0 | 25 (100%) | 0 | 3 (100%) | 0 |
| Q2.1 | 227 | 44 (19%) | 3 (7%) | 39 (89%) | 2 (5%) | 1 (100%) | 0 |
| Q2.2 | 199 | 52 (26%) | 0 | 49 (94%) | 3 (6%) | 0 | 0 |
| Q2.3 | 54 | 14 (26%) | 0 | 14(100%) | 0 | 0 | 0 |
| Q3.1 | 23 | 5 (22%) | 0 | 5 (100%) | 0 | 1 (50%) | 1 (50%) |
| Q3.2 | 59 | 21 (55%) | 0 | 16 (71%) | 5 (29%) | 1 (100%) | 0 |
| Q3.3[a] | 10[a] | 2 (20%)[a] | 0[a] | 2 (100%)[a] | 0[a] | 0[a] | 0[a] |
| Q4.1 | 2 | 1 (50%) | 0 | 1 (100%) | 0 | 0 | 0 |
| Q4.2 | 82 | 0 | 0 | 0 | 0 | 0 | 0 |
| Q4.3 | 20 | 2 (10%) | 0 | 2 (100%) | 0 | 0 | 0 |
| Total | 893 | 172 (19 %) | 3 (2%) | 159 (92%) | 10 (6%) | 6 (86%) | 1 (14%) |

**Notes.**
[a]Papers in Spanish excluded.

Because the focus of this paper lies on the assumptions of linear regression, only linear regression model assumptions were examined in the literature review. Consequently, papers that analysed data by means of other types of regression, such as latent factor models, logistic regression, and proportional hazards models (Cox regression), were not inspected for assumption checking. When a paper used a regression model other than linear regression, and without mentioning that linear regression was alternatively considered for data analysis it was classified as 'No Model of Interest'.

## RESULTS

The results of the systematic literature review are displayed in Tables 3–5 which display the number of occurrences of different classifications for the selected journals. In the online Supplemental Information 1 we indicate for all of the 893 individual papers studied into which category they fall.

Table 3 shows the findings for all journals with the 12 different classification rubrics summarized into seven different columns. The three columns entitled 'Dealing with assumptions' list the number of different types of regression papers in a specific journal and shows the proportional amount of this type in relation to the complete number of regression articles in that journal. The two columns for 'No regression' list the number of papers which did not use a linear regression model and included in their method sections to

**Table 4** **Breakdown of the types of mistakes that were observed.** Only Journals with flawed models are listed. Categorizations are mutually exclusive and exhaustive. Journals are referred by the labels assigned in Table 1.

| Journal | Articles with flawed linear regression model (*rub. 8–12*) | *Tested normality of X but not of residuals (rub. 8)* | *Tested normality of Y but not of residuals (rub. 9)* | *Assuming normally distributed variables but did not indicate if X or Y or both (rub. 10)* | *Tested normality of X and of Y but not of residuals (rub. 11)* | Other misconceptions (*rub. 12*) |
|---|---|---|---|---|---|---|
| Q2.1 | 2 | 0 | 0 | | 0 | 2 (100%) |
| Q2.2 | 3 | 2 (67%) | 0 | 0 | 0 | 1 (33%) |
| Q3.2 | 5 | 4 (80%) | 1 (20%) | 0 | 0 | 0 |
| Total | 10 | 6 (60%) | 1 (10%) | 0 | 0 | 3 (30%) |

**Table 5** **Breakdown of the different types of 'Unclear' classifications.** Only Journals with unclear models are listed. Categorizations are mutually exclusive and exhaustive. Journals are referred by the labels assigned in Table 1.

| Journal | Papers in which handling of regression assumption was unclear (*rub. 5–7*) | Unclear | | |
|---|---|---|---|---|
| | | If the 'normality assumption' was tested on the residuals or on *X* or *Y* (*rub. 5*) | Did not test all but some correct assumptions (*rub. 6*) | No indication if any or which assumptions were tested (*rub. 7*) |
| Q1.2 | 6 | 0 | 2 (33%) | 4 (67%) |
| Q1.3 | 26 | 0 | 0 | 25 (100%) |
| Q2.1 | 39 | 4 (10%) | 5 (13%) | 30 (77%) |
| Q2.2 | 49 | 1 (2%) | 2 (4%) | 46 (94%) |
| Q2.3 | 14 | 0 | 1 (7%) | 13 (93%) |
| Q3.1 | 5 | 0 | 0 | 5 (100%) |
| Q3.2 | 16 | 0 | 0 | 16 (100%) |
| Q3.3 | 2 | 0 | 0 | 2 (100%) |
| Q4.1 | 1 | 0 | 0 | 1 (100%) |
| Q4.3 | 2 | 0 | 0 | 2 (100%) |
| Total | 159 | 5 (3%) | 10 (6%) | 144 (91%) |

have considered a linear regression analysis but decided against it on the basis of checking either correct or incorrect assumptions.

Table 4 specifies the details behind the articles which are listed in Table 3 under the column titled 'incorrectly '. This table classifies the corresponding 10 papers into Rubrics 8–12 of Table 2. It may be noted that 4% of all articles that used linear regression checked normal distributions of some variables instead of normal distribution of errors.

Table 5 specifies the details behind the column 'unclear' in Table 2; that is it classifies the 159 corresponding papers into Rubrics 5 to 7 of Table 2. Of all papers that employed regression, 92% were unclear about the assumptions of the linear regression model that were tested or were thought to be fulfilled.

## DISCUSSION

In our analysis, we studied 893 papers, representative for the work published in the field of clinical psychology, and classified the 172 papers (19.4%) which considered linear regression into three categories: those that dealt with the assumptions correctly, those that dealt with assumptions incorrectly, and those that did not specify how they dealt with assumptions.

Merely 2% of these papers were both transparent and correct in their dealing with statistical assumptions. Furthermore, in 6% of papers transparency was given but the dealing with assumptions was incorrect. *Hoekstra, Kiers & Johnson (2012)* might provide some insight into why researchers did not check assumptions. They list unfamiliarity with either the fact that the model rests on the assumption or with how to check the assumption as the top two reasons. As explained, incorrect dealing with the assumptions could lead to severe problems regarding the validity and power of the results. We hope that this manuscript creates new awareness of this issue with editors of clinical psychology journals and that this will assist in bringing down the number of publications with flawed statistical analyses.

A tremendous amount of papers that employed regression, 92% of those studied, were not clear on how they dealt with assumptions. It is not possible (not for us, nor for the reader) to judge from the text whether checks for assumption violations were performed correctly. In the group of transparent papers, the number of papers with fundamental mistakes in dealing with assumptions far outnumber the number of papers without mistakes. Thus, even though it is not possible to pinpoint an exact number to it, it would be naive to assume that only a small proportion of those 92% also deal with assumptions incorrectly.

We believe that most contemporary problems in the handling of regression methods could be counteracted by a more thorough coverage of the statistical assumption checks that were performed in order to determine the validity of the linear regression model. At the very least, transparency regarding how assumptions are approached, in line with the recommendations by *Wilkinson & Task Force on Statistical Inference (1999)*, is essential. Thus, mentioning which assumptions were checked and what diagnostic tools were used to check them under what criteria, should be a minimum requirement. Preferably, the authors should also show the results of these checks.

With transparency, the critical reader can distinguish correct approaches from incorrect ones, even if the author(s), editor(s) and referees fail to spot the flaws. These statistical checks can be given in the paper itself, but could also be provided in online Supplemental Information, a possibility most journals offer nowadays (note that none of the papers investigated in this manuscript referred to Supplemental Information for assumption checks). Thus, increased length of the manuscript does not need to be an issue. Our aspiration for an increased transparency in statistical assumption checks is in line with recent developments in psychology such as open methods (obligatory in e.g., the APA-journal Archives of Scientific Psychology) and open data (either published as online Supplemental Information with a paper, or through special journals like Journal of Open Psychology Data). With open data, sceptical scientists can re-do the analyses and check

assumptions for themselves. Enforcing, or at least strongly encouraging, transparency can even have beneficial effects to the level of publications in the respective journal (*Wicherts, Bakker & Molenaar, 2011*). Even if publishing the data does not have a direct beneficial effect on the quality of work, it will be useful as it provides the sceptical reader with the required information to perform the assumption checks and thus the possibility to check the credibility of the published work.

It is difficult to establish whether high ranking journals deal with assumptions more adequately than lower ranking journals. Even though the results in Table 5 indicate that higher ranked journals were more likely to test at least a few assumptions compared to lower ranked journals; the results do mainly show that there is great variability between journals regarding the number of papers with applied regression models they publish: two journals published no papers in 2013 that employed linear regression, and five journals published six or fewer of these papers. Because two of the three inspected Q1 journals are review journals they predominantly employed meta-regression, a special type of regression useful for conducting meta-analyses, and only rarely linear regression, it should be pointed out that of the 15 papers that used meta-regressions in our Q1.2 eleven tested at least some of the required assumptions (that is 73% of meta-regression papers were checked correctly for statistical assumptions). We believe that for these papers the percentage is much better than the overall percentage of 2% for applied regression papers, because meta-analyses are usually carried out by a team of authors including at least one statistician or psychometrician.

We have limited our literature review to papers employing linear regression models, in order to keep the study feasible. We suspect that similar findings would arise when studying other classes of statistical models. Furthermore, we have also limited the review to papers published in the field of clinical psychology; however we suspect that similar problems occur—albeit possibly in different proportions—in all areas of applied psychological research. Thus, our suggestions with respect to increased transparency and better evaluation of the employed methodology should be relevant for a wider range of papers than those studied here. Because our categorization of papers is reasonably straightforward, only one author conducted most of the review. While our rubrics allow objective classifications we cannot preclude a few single accidental misclassifications. However, possible misclassification should be minimal at most and can therefore be expected to not have skewed the overall results that are based on a large number of papers. Thus, despite this limitation we are confident in the overall results. For future research, it would be interesting to do a similar literature review based on either alternative techniques or on another field of application. Furthermore, more research is needed in understanding the reasons that underlie why researchers frequently do not check assumptions.

One of the consequences of the lack of reporting of assumption checks is that many published findings in clinical psychology are underestimating the uncertainty in their claims. For instance, reported confidence intervals in the literature describe the uncertainty surrounding the parameter, if the OLS-assumptions are met. The uncertainty of the validity of the assumptions should lead to wider confidence intervals, in general. For future research, it would be an interesting puzzle to assess the magnitude of this added uncertainty.

To summarise, in order to prevent the observed problems that were outlined above we suggest a more transparent methodological reporting. Research should cover which assumption checks were carried out. Furthermore, it should be mentioned if alternative statistical models have been considered and why they were not employed, if so. This will be a necessity for future research articles in order to be able to detect and prevent errors related to widespread misconceptions but also to remove doubt from articles with an actual immaculate data analysis.

### Funding
The authors received no funding for this work.

### Competing Interests
The authors declare there are no competing interests.

### Author Contributions
- Anja F. Ernst and Casper J. Albers conceived and designed the experiments, analyzed the data, wrote the paper, prepared figures and/or tables, reviewed drafts of the paper.

### Data Availability
The raw data has been supplied as a Supplementary File.

### Supplemental Information
Supplemental information for this article can be found online at http://dx.doi.org/10.7717/peerj.3323#supplemental-information.

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
