# Peer review of "Regression assumptions in clinical psychology research practice—a systematic review of common misconceptions"

_PeerJ, doi:10.7717/peerj.3323_

## Round 0.1 · original submission · Major Revisions

I received two very detailed reviews that outline both minor and more substantial questions, wording suggestions, and conceptual issues to address. One reviewer voiced a lot of positivity about the manuscript; the second did not identify any issues that would prevent publication after revision. Both reviewers provided very detailed, comprehensive, and reasonable commentary.

I would suggest that (1) you deal with and address all minor revisions and clarifications the reviewers made; (2) for the more major substantial suggestions made by Reviewer 1, address them or justify why you feel it is not necessary in the context of PeerJ's editorial guidelines.

In reviewing the paper and the comments, I think that one issue that I and the reviewers would like to have clear throughout is something about the consequences of different assumptions. As reviewer 1 suggests, some of the assumptions (i.e., normality) are not necessary for unbiased estimation of linear terms; probably all you need is an error distribution with finite variance that is has a mean of 0 and is symmetric. But inferential tests are based on the estimated error distribution, and mis-specifying the error distribution here could prevent reasonable inference (and impact type I and type II errors).

A second issue that I don't think was addressed by the reviewers relates to the typical 'best practices' for testing these assumptions. I think the contribution of the paper, in general, could be improved by including a section in the discussion on best practices, perhaps with comments on consequences of violations.

Specifically, consider the case of normality, I know of four basic methods people use for testing normality: looking at histograms of residuals (is is sort of bell-shaped and symmetric); looking at normal q-q plots (is is mostly flat?); doing a shapiro-wilks or another similar special-purpose test for normality, or doing something like a kolmogorov-smirnov test against a normal distribution 'by hand', which might amount to the same thing. Nevertheless, shapiro's test will often fail on something like likert-scale responses, even if the distribution is symmetric and the qq-plot is flat, because the residuals are quantized. In point of fact, almost all dependent measures used in clinical psychology logically and necessarily violate normality: because they will typically not have support on the continuous real number line; because most measures will have minimum (often 0) and a maximum, and because oftentimes measured only on an ordinal or integer scale.

Similarly, in repeated-measures ANOVA, people typically look at the Mauchsley's 'sphericity' test that SPSS routinely report, partially as a check for certain types of independence, and probably heteroscedascity. Certain modern regression approaches for mixed effects modeling (e.g., lmer) will estimate correlations between parameters, which I think is another check on independence.

What are the gold-standard or best practices for checking these assumptions, or at least what would have constituted reasonable checks in this analysis? I think such a description could better document what you considered satisfactory checks, but could also provide guidance to future researchers about what is normally done, or what could be done.

·

Basic reporting

Please see the attached pdf review.

Experimental design

Please see the attached pdf review.

Validity of the findings

Please see the attached pdf review.

Additional comments

Please see the attached pdf review.

·

Basic reporting

• Page 5. Lines 5-10: Can be stated more concisely: “Most commonly, researchers incorrectly assume that Xi, or both Xi and Y, should be normally distributed instead of the residuals. Osborne and Waters (2002), which has been viewed online over 360,000 times at the time of writing, make this mistake (cf. Williams, Grajales & Kurkiewics, 2013), demonstrating how widespread this misconception really is.”
• Page 9. Line 9: I don’t understand the intention of “including those that had already been published earlier as well.” Do you mean that articles published before 2013 were also included?
• Page 9. Lines 11-12: The phrase “also not in the section ‘No Model of Interest’” is awkward here. I don’t think it’s necessary because you make it clear that these types of documents were not included in your literature search at all.
• Page 9. Lines 13-14: Potential supplemental material, but please provide a citation/reference for PsycARTICLES and other websites.
• Page 12. Line 8: Say which types of mistakes were made that are warned for in undergraduate statistics textbooks. Citation?
• Page 16. Lines 9 & 12: Hoekstra references are not aligned properly.
• Page 19. Table 3: The use of “col.” is confusing because Table 2 is not organized into columns. “Row” or “rubric” may be more appropriate. If you list column/rubric numbers in this table then please do so in Tables 4 & 5. Break the underline between "Dealing with assumptions" and "No regression" so that they are more easily seen as separate groups.
• Page 22. Figure 2: The word “scatterplot” implies there will be points in the plot but there are none.

Experimental design

• Page 7. Line 18: The use of the phrase “for valid reasons” here seems to imply that violations of linearity or independence are valid reasons for rejecting linear regression but violations of normality or homoscedasticity are not, which undermines your earlier argument. If some of the assumptions are softer requirements than others, please acknowledge this somewhere in the introduction and consider discussing this with respect to your results later in the document.
• Page 9. Lines 1-5: Be clear on the exact criterions used in filtering the journals on SCImago. I had trouble choosing the appropriate region/country when attempting to corroborate your list of journals.
• Page 9. Line 7: Is there any reason to think that linear regression assumption checking is evenly distributed within quantiles? I.e. is there any way that this selection method happens to choose unrepresentative journals?

Validity of the findings

• Page 6. Lines 13-14: If the predictor variable (Y) is normally distributed, then won’t the residuals be normally distributed as well? In this case, it would not be incorrect to accept the regression assumption of normality based on normality of variables. Then rubric items 9 & 10 should be dropped, which I believe changes the categorization of one article.
• Page 7. Lines 12-14: Although the non-parametric test may be less powerful than linear regression, would the non-parametric test still be a valid approach?
• Page 9. Lines 6-7: Do these 12 journals have their own guidelines for authors when reporting statistical tests/assumptions?

Additional comments

Other comments and suggestions:
• Page 3. Line 12: “called the simple linear regression model.”
• Page 4. Lines 19-21: awkward wording/grammar. Try “Figure 1 displays what an example scatterplot may look like for three levels of compliance with each assumption.”
• Page 4. Line 12: I think this is supposed to be Figure 1 not 2. Also need comma after “QQ-plots”
• Page 4. Line 14: I think APA requires “e.g.” to be used within parentheses.
• Pages 3-4: I understand the intention behind underlining the emphasized elements for each assumption, but I’m not sure it’s needed.
• Page 5. Lines 4-5: Be explicit: “most of which concern the assumptions of normality and homoscedasticity.”
• Page 5. Line 5: missing comma “assume that Xi, or both Xi and Y,”
• Page 5. Lines 12-15: Because you don’t really revisit this particular example later in the text, I would replace the colon and words that follow with a parenthetical “e.g.”
• Page 5. Lines 17-18: Strike “so” and place the subsequent “i.e.” clause in parentheses.
• Page 5. Line 20: Missing “a” before "label".
• Page 7. Line 2: Strike “However” and begin third sentence with “In any case,”.
• Page 9. Line 21: Try “which recommend a researcher carefully examine the data”.
• Page 12. Line 11: No comma after “assumption”.
• Page 12. Line 12: No comma after “assumptions”.
• Page 12. Line 15: Subject-verb agreement. Should be "will assist" or "assists" but not "will assists".
• Page 12. Line 21: Should be “92% also deal with” not “92% is also dealing with”.
• Page 13. Line 1: Missing a period between sentences.
• Page 14. Line 21: I suggest either strike "the" or list the widespread misconceptions.

---

## Round 0.2 · Minor Revisions

The revised ms was reviewed by myself and one expert reviewer, who has recommended accept but made a lot of detailed comments and suggestions. I am in agreement that the the paper should be accepted, but I'd like to give you a chance to deal with the reviewers detailed comments to improve the ms. I would like to request that you submit a clean revision, incorporating the comments made by the reviewer that you feel are justified. Given the reviewer suggested that a careful proofreading is probably needed, I'll do one more careful reading of this next revision before accepting fully.

·

Basic reporting

See general comments

Experimental design

See general comments

Validity of the findings

See general comments

Additional comments

Well done on producing such thorough revisions. I did provide quite a long list of suggested changes, and you've done a very good job at incorporating these suggestions.

At this stage I recommend that the article be accepted for publication. I have attached a copy of the manuscript with some minor suggestions added via tracked changes and bubble comments; these suggestions focus only on the clarity of the writing rather than substantive scientific issues. They are *not* compulsory amendments, just gentle suggestions for you to apply if they find them helpful. (Note: My comments and tracked changes are the ones in light blue).

---

## Round 0.3 · Minor Revisions

Thank you for the revision. As mentioned in the previous decision, I wanted to make one final pass over the manuscript once you had dealt with reviewer's comments, before accepting completely. Consequently, you can consider this 'accepted pending minor changes'.

I have edited with track changes (and I'll try to send the .docx to you directly, as i added some 'comments' which do not appear on the pdf). I've made a number of wording suggestions that either fix grammatical errors or improve readability in my opinion. I will let you judge which of them you want to accept--although most of these are minor grammatical issues or typos. My only 'requirement' is dealing with the inter-coder reliability issue in the method section (see below).


My edits appear as tan/yellow track changes. There are three points where I have suggested a more substantial sentence-level change:


* Line 82: I suggested new wording that makes it read better, moving the download count to the footnote. In addition, although this is a nitpick--the original wording sort of blamed the authors for making the mistake, whereas my suggestion is to blame the paper, which is less of an ad hominem.

* Line 196: this comment is self-explanatory, but I think that using a standard regression (especially ANOVA) for a repeated-measures design is one of the most common regression errors people make, and this is specifically related to independence. If you agree, feel free to use this or put it in your own words.

*Around line 255, you should describe methods used to ensure reliability of coding. I know that this was discussed briefly in the results and also appears in the additional notes section, but it belongs in the method section. I had even recalled that you had reported something about this and was looking for it but could not find it because it did not appear in the methods and did not match any of the obvious search terms I looked for (reliability, inter-rater, kappa, etc., consensus, etc.)
Complete agreement on a subset should be sufficient and so prevent criticism about not having multiple independent coders and a kappa value to report. Having multiple independent raters is actually a publication requirement in some clinical journals, and I have had papers rejected without review because a classification of patient status was done by just one physician. Clinical researchers--who are being held up to a magnifying glass here-- will be very sensitive to this, and it would be ironic if a paper criticizing them for failing to describe methods made a similar mistake by not fully describing the methods in a way future researchers could easily find!

Thanks,
Shane

---

## Round 0.4 · accepted · Accept

Thank you for the revisions. Accepted!